# Reproducibility and Scientific Integrity of Big Data Research in Urban Public Health and Digital Epidemiology: A Call to Action

**DOI:** 10.3390/ijerph20021473

**Published:** 2023-01-13

**Authors:** Ana Cecilia Quiroga Gutierrez, Daniel J. Lindegger, Ala Taji Heravi, Thomas Stojanov, Martin Sykora, Suzanne Elayan, Stephen J. Mooney, John A. Naslund, Marta Fadda, Oliver Gruebner

**Affiliations:** 1Department of Health Sciences and Medicine, University of Lucerne, 6002 Luzern, Switzerland; 2Institute of Global Health, University of Geneva, 1211 Geneva, Switzerland; 3CLEAR Methods Center, Department of Clinical Research, Division of Clinical Epidemiology, University Hospital Basel and University of Basel, 4031 Basel, Switzerland; 4Department of Orthopaedic Surgery and Traumatology, University Hospital of Basel, 4031 Basel, Switzerland; 5School of Business and Economics, Centre for Information Management, Loughborough University, Loughborough LE11 3TU, UK; 6Department of Epidemiology, University of Washington, Seattle, WA 98195, USA; 7Department of Global Health and Social Medicine, Harvard Medical School, Boston, MA 02115, USA; 8Institute of Public Health, Università Della Svizzera Italiana, 6900 Lugano, Switzerland; 9Epidemiology, Biostatistics and Prevention Institute, University of Zurich, 8001 Zurich, Switzerland; 10Department of Geography, University of Zurich, 8057 Zurich, Switzerland

**Keywords:** reproducibility, big data, digital epidemiology, urban public health

## Abstract

The emergence of big data science presents a unique opportunity to improve public-health research practices. Because working with big data is inherently complex, big data research must be clear and transparent to avoid reproducibility issues and positively impact population health. Timely implementation of solution-focused approaches is critical as new data sources and methods take root in public-health research, including urban public health and digital epidemiology. This commentary highlights methodological and analytic approaches that can reduce research waste and improve the reproducibility and replicability of big data research in public health. The recommendations described in this commentary, including a focus on practices, publication norms, and education, are neither exhaustive nor unique to big data, but, nonetheless, implementing them can broadly improve public-health research. Clearly defined and openly shared guidelines will not only improve the quality of current research practices but also initiate change at multiple levels: the individual level, the institutional level, and the international level.

## 1. Introduction

Research comprises “creative and systematic work undertaken in order to increase the stock of knowledge” [1,2]. Research waste, or research whose results offer no social benefit [3], was characterized in a landmark series of papers in the Lancet in 2014 [4,5]. The underlying drivers of research waste range from methodological weaknesses in specific studies to systemic shortcomings within the broader research ecosystem, notably including a reward system that incentivises quantity over quality and incentivizes exploring new hypotheses over confirming old ones [4,5,6,7,8].

Published research that cannot be reproduced is wasteful due to doubts about its quality and reliability. Lack of reproducibility is a concern in all scientific research, and it is especially significant in the field of public health, where research aims to improve treatment practices and policies that have widespread implications. In this commentary, we highlight the urgency of improving norms for reproducibility and scientific integrity in urban public health and digital epidemiology and discuss potential approaches. We first discuss some examples of big data sources and their uses in urban public health, digital epidemiology, and other fields, and consider the limitations with the use of big data. We then provide an overview of relevant solutions to address the key challenges to reproducibility and scientific integrity. Finally, we consider some of their expected outcomes, challenges, and implications.

Unreliable research findings also represent a serious challenge in public-health research. While the peer-review process is designed to ensure the quality and integrity of scientific publications, the implementation of peer review varies between journals and disciplines and does not guarantee that the data used are properly collected or employed. As a result, reproducibility remains a challenge. This is also true in the context of the emerging field of big data science. This is largely driven by the characteristics of big data, such as their volume, variety, and velocity, as well as the novelty and excitement surrounding new data science methods, lack of established reporting standards, and a nascent field that continues to change rapidly in parallel to the development of new technological and analytic innovations. Recent reports have uncovered that most research is not reproducible, with findings casting doubt on the scientific integrity of much of the current research landscape [6,9,10,11,12]. At the bottom of this reproducibility crisis lies growing pressure to publish not only novel, but more importantly, statistically significant results at an accelerated pace [13,14], increasing the use of low standards of evidence and disregarding pragmatic metrics, such as clinical or practical significance [15]. Consequently, the credibility of scientific findings is decreasing, potentially leading to cynicism or reputational damage to the research community [16,17]. Addressing the reproducibility crisis is not only one step towards restoring the public’s trust in scientific research, but also a necessary foundation for future research, as well as guiding evidence-based public-health initiatives and policies [18], facilitating translation and implementation of research findings [19,20], and accelerating scientific discovery [21].

While failure to fully document the scientific steps taken in a research project is a fundamental challenge across all research, big data research is additionally burdened by the technical and computational complexities of handling and analysing large datasets. The challenge of ensuring computational capacity, including memory and processing power, to handle the data, as well as statistical and subject matter expertise accounting for data heterogeneity, can lead to reproducibility issues at a more pragmatic level. For example, large datasets derived from social media platforms require data analysis infrastructure, software, and technical skills, which are not always accessible to every research team [22,23]. Likewise, studies involving big data create new methodological challenges for researchers as the complexity for analysis and reporting increases [24]. This complexity not only requires sophisticated statistical skills but also new guidelines that define how data should be processed, shared, and communicated to guarantee reproducibility and maintain scientific integrity, while protecting private and sensitive information. Some of these challenges lie beyond the abilities and limitations of individual researchers and even institutions, requiring cultural and systematic changes to improve not only the reproducibility but also transparency and quality of big data research in public health.

Importantly, through concerted efforts and collaboration across disciplines, there are opportunities to systematically identify and address this reproducibility crisis and to specifically apply these approaches to big data research in public health. Below, we discuss methodological and analytical approaches to address the previously discussed issues, reduce waste, and improve the reproducibility and replicability of big data research in public health.

Specifically, we focus on approaches to improve reproducibility, which is distinct from replicability. While both are important with regards to research ethics, replicability is about “obtaining consistent results across studies aimed at answering the same scientific question, each of which has obtained its own data”, whereas reproducibility refers to “obtaining consistent results using the same input data’ computational steps, methods and code, and conditions of analysis” [25]. Though we mention “reproducibility” throughout this commentary, some of the arguments presented may apply to replicability as well. This is particularly true when it comes to transparency when reporting sampling, data collection, aggregation, inference methods, and study context; these affect both replication and reproduction [26].

## 2. Big Data Sources and Uses in Urban Public Health and Digital Epidemiology

Big data, as well as relevant methods and analytical approaches, have gained increasing popularity in recent years. This is reflected in the growing number of publications and research studies that have implemented big data methods across a variety of fields and sectors, such as manufacturing [27], supply-chain management [28], sports [29], education [30], and public health [31].

Public health, including urban health and epidemiological research, is a field where studies increasingly rely on big data methods, such as in the relatively new field of digital epidemiology [32]. The use of big data in public-health research is often characterized by the ‘3Vs’: variety in types of data as well as purposes; volume or amount of data; and velocity, referring to the speed at which the data are generated [33]. Because large datasets can invariably produce statistically significant findings but systematic biases are unaffected by data scale, big data studies are at greater risk of producing inaccurate results [34,35,36,37].

Big data sources that are used or could be potentially used in fields, such as urban public health and digital epidemiology, can be divided into two main categories. First, those that are collected or generated with health as a main focus and, second, those that are generated out of this scope but that can be associated with or impact public health (Figure 1) [32].

Data sources generated within the context of public health include large datasets captured within health systems or government health services at the population level, such as the case of Electronic Health Records (EHRs), Electronic Medical Records (EMRs), or personal health records (PHRs) [38]. Other examples include pharmacy and insurance records, omics data, as well as data collected by sensors and devices that are part of the internet of things (IoT) and are used for health purposes, ranging from smart continuous glucose monitors (CGMs) [39] to activity and sleep trackers.

In contrast, big data sources generated outside the public-health scope are virtually unlimited and ever-growing, covering virtually all domains in society. As a result, we will focus on selected and non-conclusive examples to illustrate and exemplify the diverse sources of big data that are used or could potentially be used in urban public health and digital epidemiology. Notably, social media have become an important source of big data used for research in different fields, including digital epidemiology. Twitter data have proven to be useful for collecting public-health information, for example, to measure mental health in different patient subgroups [40]. Examples of big data collected on Twitter that can be used in the context of public-health research are the Harvard CGA Geotweet Archive [41] or the University of Zurich Social Media Mental Health Surveillance project with their Geotweet Repository for the wider European Region [42]. Other initiatives, such as the SoBigData Research Infrastructure (RI), aim to foster reproducible and ethical research through the creation of a ‘Social Mining & Big Data Ecosystem’, allowing for the comparison, re-use, and integration of big data, methods, and services into research [22].

Cities increasingly use technological solutions, including IoT and multiple sensors, to monitor the urban environment, transitioning into Smart Cities with the objective of improving citizens’ quality of life [43,44]. Data stemming from Smart City applications have been used, for example, to predict air quality [45], analyse transportation to improve road safety [46], and have the potential to inform urban planning and policy design to build healthier and more sustainable cities [47].

Data mining techniques also allow for large datasets to be used in the context of urban public health and digital epidemiology. For example, a project using administrative data and data mining techniques in El Salvador identified anomalous spatiotemporal patterns of sexual violence and informed ways in which such analysis can be conducted in real time to allow for local law enforcement agencies and policy makers to respond appropriately [48,49]. Other large-dataset sources, such as transaction data [50], have been used to investigate the effect of sugar taxes [51] or labelling [52] on the consumption of healthy or unhealthy beverages and food products, which can eventually help model their potential impact on health outcomes.

## 3. Approaches to Improving Reproducibility and Scientific Integrity

Big data science has brought on new challenges, to which the scientific community needs to adapt by applying adequate ethical, methodological, and technological frameworks to cope with the increasing amount of data produced [53]. As a result, the timely adoption of approaches to address reproducibility and scientific integrity issues is imperative to ensure quality research and outcomes. A timely adoption is relevant not only for the scientific community but also for the general public that can potentially benefit from knowledge and advancements resulting from the use of big data research. This is particularly important in the context of urban public health and digital epidemiology, as the use of big data in these fields can help answer highly relevant and pressing descriptive (what is happening), predictive (what could happen), and prescriptive (why it happened) research questions [54]. A brief summary of the main points discussed in this section can be found in Figure 2. We divide our proposed solutions in this commentary into three main domains: (1) good research practice, (2) scientific communication and publication, and (3) education.

### 3.1. Good Research Practice

Practices, such as pre-registration of protocols, predefining research questions and hypotheses, publicly sharing data analysis plans, and communicating through reporting guidelines, can improve the quality and reliability of research and results [55,56]. For experimental studies, clear and complete reporting and documentation are essential to allow for reproduction. Observational studies can also be registered on well-established registries, such as on clinicaltrials.gov. Importantly, pre-registration does not preclude publishing exploratory results; rather, it encourages such endeavours to be explicitly described as exploratory, with defined hypotheses and expected outcomes, which is appropriate [35,37].

Lack of data access is another key challenge to reproducibility. Adoption of open-science practices, including sharing of data and code, represents a partial solution to this issue [57,58], acknowledging that not all data can be shared openly owing to privacy concerns. Similarly, transparent descriptions of data collection and analytic methods are necessary for reproduction [59]. For example, in the analysis of human mobility, which has applications in a wide range of fields, including public health and digital epidemiology [60,61], the inference of ‘meaningful’ locations [62] from mobility data has been approached with a multitude of methods, some of which lack sufficient documentation. Whereas a research project using an undocumented method to identify subject homes cannot be reproduced, a project using Chen and Poorthius’s [63] R package ‘homelocator‘, which is open source and freely available, could be.

Likewise, a case could be made to collaboratively share big data within research networks and IT infrastructures. An example of a project tackling this issue in the context of public health is currently being developed by the Swiss Learning Health System (SLHS), focusing on the design and implementation of a metadata repository with the goal of developing Integrated Health Information Systems (HISs) in the Swiss context [64,65]. The implementation of such repositories and data-management systems allows for retrieval of and access to information; nevertheless, as information systems develop, new challenges arise, particularly when it comes to infrastructure as well as legal and ethical issues, such as data privacy. Solutions are currently in development; it is likely that decentralised data architectures based on blockchain will play an important role in integrated care and health information models [66]. We briefly expand on this topic in the Anticipated Challenges section below.

The adoption of appropriate big data handling techniques and analytical methods is also important to ensure the findability, accessibility, interoperability, and reusability (FAIR) [67] of both data and research outcomes [68]. Such characteristics allow for different stakeholders to use and reuse data and research outcomes for further research, replication, or even implementation purposes.

Complete and standardised reporting of aspects discussed in this section, for instance, in Reproducibility Network Groups, allows for meta-research and meta-analyses, the detection and minimization of publication bias, and the evaluation of the adherence of researchers to guidelines focused on ensuring scientific integrity. The use of checklists by individual researchers, research groups, departments, or even institutions can motivate the implementation of good research practices as well as clear and transparent reporting, ultimately improving research integrity [69]. Such checklists can serve as training tools for younger researchers, as well as offer practice guidelines to ensure quality research.

Senior researchers and research institutions are vital when it comes to tackling these challenges as well. The adoption of principles for research conduct, such as the Hong Kong principles, can help minimise the use of questionable research practices [70]. These principles are to: (1) assess responsible research practices; (2) value complete reporting; (3) reward the practice of open science; (4) acknowledge a broad range of research activities; and (5) recognise essential other tasks, such as peer review and mentoring [71]. The promotion of these principles by mentors and institutions is a cornerstone of good research practices for younger researchers.

### 3.2. Scientific Communication

Scientific communication, not only between researchers but also between institutions, should be promoted. Recently, requirements for researchers to make data public or open source have grown popular among journals and major funding agencies in the US, Europe, and globally; this is an important catalyst for open science and addressing issues such as reproducibility [72].

Likewise, publication and sharing of protocols, data, code, analysis, and tools are important. This not only facilitates reproducibility but also promotes openness and transparency [73]. For example, the Journal of Memory and Language adopted a mandatory data-sharing policy in 2019. An evaluation of this policy implementation found that data sharing increased more than 50% and the strongest predictor for reproducibility was the sharing of analysis code, increasing the probability of reproducibility by 40% [57]. Such practices are also fostered by the creation and use of infrastructure, such as the aforementioned SoBigData, and reproducibility network groups, such as the Swiss Reproducibility Network, a peer-lead group that aims to improve both replicability and reproducibility [74], improve communication, collaboration, and encourage the use of rigorous research practices.

When publishing or communicating their work, researchers should also keep in mind that transparency regarding whether studies are exploratory (hypothesis forming) or confirmatory (hypothesis testing) is important to distinguish from testing newly formed hypotheses and the testing of existing ones [75]; this is particularly important for informing future research. Journal reviewers and referees should also motivate researchers to accurately report this.

Similarly, when publishing results, the quality, impact, and relevance of a publication should be valued more than scores, such as the impact factor, to avoid “publishing for numbers” [76]. This would, of course, require a shift in the priorities and views shared within the research community and may be a challenging change to effect.

Academic editors can also play an important role by avoiding practices, such as ‘cherry-picking’ publications, either because of statistical significance of results or notoriety of the authors. Instead, practical significance, topic relevance, and replication studies should be important factors to consider, as well as valuing the reporting of negative results. It is important to acknowledge, though, that scientific publication structures face an important number of challenges that hinder the implementation of these practices. Some of these points are mentioned in the Challenges section that follows.

### 3.3. Education

Academic institutions have the responsibility to educate researchers in an integral way, covering not only the correct implementation of methodological approaches and appropriate reporting but also how to conduct research in an ethical way.

First, competence and capacity building should be addressed explicitly through courses, workshops, and competence-building programs aimed at developing technical skills, good research practices, and adequate application of methods and analytical tools. Other activities such as journal clubs can allow researchers to exchange and become familiar with different methodologies, stay up to date with current knowledge and ongoing research, and develop critical thinking skills [77,78], while fostering a mindset for continuous growth and improvement.

Second, by incorporating practice-based education, particularly with research groups that already adhere to best practices, such as the Hong Kong principles, institutions can foster norms valuing reproducibility implicitly as an aspect of researcher education.

## 4. Expected Outcomes

Ideally, successful implementation of the approaches proposed in Figure 2, and the methodological and analytical approaches, such as the standardised protocols that were suggested by Simera et al. [55] and the Equator Network reporting guidelines [79], can potentially lead to a cultural shift in the research community. This, in turn, can enhance transparency and the quality of public-health research using big data by fostering interdisciplinary programs and worldwide cooperation among different health-related stakeholders, such as researchers, policy makers, clinicians, providers, and the public. Improving research quality can lead to greater value and reliability, while decreasing research waste, thus, improving the cost–value ratio and trust between stakeholders [80,81], and as previously stated, facilitating translation and implementation of research findings [18].

Just in the way replicability is fundamental in engineering to create functioning and reliable products or systems, replicability is also necessary for modelling and simulation in the fields of urban public health and digital epidemiology [82]. Simulation approaches built upon reproducible research allow for the construction of accurate prediction models with important implications for healthcare [83] and public health [84]. In the same way, reproduction and replication of results for model validation are essential [85,86,87].

The importance of reducing research waste and ensuring the value of health-related research is reflected in the existence of initiatives, such as the AllTrials Campaign, EQUATOR (enhancing the quality and transparency of health research), and EVBRES (evidence-based research), which promote protocol registration, full methods, and result reporting, and new studies that build on an existing evidence base [79,88,89,90].

Changes in editorial policies and practices can improve critical reflection on research quality by the authors. Having researchers, editors, and reviewers use guidelines [91], such as ARRIVE [92] in the case of pre-clinical animal studies or STROBE [93] for observational studies in epidemiology, can significantly improve reporting and transparency. For example, an observational cohort study analysing the effects of a change in the editorial policy of *Nature*, which introduced a checklist for manuscript preparation, demonstrated that reporting risk of bias improved substantially as a consequence [94].

A valuable outcome of adopting open science approaches that could result in improved communication, shared infrastructure, open data, and collaboration between researchers and even institutions is the implementation of competitions, challenges, or even ‘hackathons’. These events are already common among other disciplines, such as computer science, the digital tech sector, and social media research, and are becoming increasingly popular in areas related to public health. Some examples include the Big Data Hackathon San Diego, where the theme for 2022 was ‘Tackling Real-world Challenges in Healthcare’ [95], and the Yale CBIT Healthcare Hackathon of 2021, which aimed to build solutions to challenges faced in healthcare [96]. In addition to tackling issues in innovative ways, hackathons and other similar open initiatives invite the public to learn about and engage with science [97] and can be powerful tools for engaging diverse stakeholders and training beyond the classroom [98].

## 5. Anticipated Challenges

While the implementation of the approaches discussed (Figure 2) will ideally translate to a significant reduction in research waste and improvement in scientific research through standardization and transparency, there are also substantial challenges to consider (Figure 3).

First, not all researchers have the adequate resources or opportunities to take advantage of new data that can be used to prevent, monitor, and improve population health. Early career researchers in low-resource settings may be at a particular disadvantage. Among these researchers, barriers to access and adequately using big data may not only be financial, when funding is not available, but also technical, when the knowledge and tools required are not available.

Similarly, events and activities among young researchers can facilitate technical development, networking, and knowledge acquisition, ultimately improving research quality and outcomes. Those who live in environments with limited resources, who are physically isolated, or have limited mobility may not have access to these opportunities. It might be possible to overcome some of these limitations with accessible digital solutions.

Much needed shifts in the research and publishing culture that currently enable Questionable Research Practices (QRPs), such as cherry picking (presenting favourable evidence or results while hiding unfavourable ones), p-hacking (misusing data through relentless analysis in order to obtain statistically significant results), HARKing (Hypothesizing After the Results are Known), among others [59,99,100]. To overcome these particular challenges embedded in modern day research, it is necessary to educate researchers about the scope of misconduct, create structures to avoid it from happening, and scrutinize cases in which these instances may be apparent to determine the actual motive [101].

Conventional data storing and handling strategies are not sufficient when working with big data, as these often impose additional monetary and computational costs. Some solutions are available to tackle these issues, such as cloud computing and platforms that allow end users to access shared resources over the internet [102]; Data Lakes, consisting of centralized repositories that allow for data storage and analysis [103]; and Data Mesh, a platform architecture that distributes data among several nodes [104]. Unfortunately, these solutions are not always easily accessible. Additionally, use of these platforms has given rise to important debates concerning issues, such as data governance and security [105].

The use of big data, and especially the use of personal and health information, raises privacy issues. The availability of personal and health information that results from the digital transformation represents a constant challenge when it comes to drawing a line between public and private, sensitive and non-sensitive information, and adherence to ethical research practices [106].

Ethical concerns are not limited to privacy; while big data entails the use of increasingly complex analytical methods that require expertise in order to deal with noise and uncertainty, there are several additional factors that may affect the accuracy of research results [107]. For example, when using machine learning approaches to analyse big data, methods should be cautiously chosen to avoid issues, such as undesired data-driven variable selection, algorithmic biases, and overfitting the analytic models [108]. Complexity increases the need for collaboration, which makes “team science” and other collaborative problem-solving events (such as Hackathons) increasingly popular. This leads to new requirements to adequately value and acknowledge contributorship [109].

Because statistical methods are becoming increasingly complex and the quantity of data is becoming greater, the number of scientific publications is also increasing, making it challenging for already-flawed peer-review systems to keep up by providing high-quality reviews to more and more complex research. Currently, there are mainly four expectations from peer-review processes: (i) assure quality and accuracy of research, (ii) establish a hierarchy of published work, (iii) provide fair and equal opportunities, and (iv) assure fraud-free research [110]; however, it is not certain whether current peer-review procedures achieve or are capable of delivering such expectations. Some solutions have been proposed to address these issues, such as the automation of peer-review processes [111] and the implementation of open review guidelines [112,113,114].

## 6. Conclusions

Big data research presents a unique opportunity for a cultural shift in the way public-health research is conducted today. At the same time, big data use will only result in a beneficial impact to the field if used adequately, taking the appropriate measures so that their full potential can be harnessed. The inherent complexity in working with large data quantities requires a clear and transparent framework at multiple levels, ranging from protocols and methods used by individual scientists to institution’s guiding dogma, research, and publishing practices.

The solutions summarized in this commentary are aimed at enhancing results, reproducibility, and scientific integrity; however, we acknowledge that these solutions are not exhaustive and there may be many other promising approaches to improve the integrity of big data research as it applies to public health. The solutions described in this commentary are in line with “a manifesto for reproducible science” published in the Nature Human Behavior journal [101]. Importantly, reproducibility is only of value if the findings are expected to have an important impact on science, health, and society. Reproducibility of results is highly relevant for funding agencies and governments, who often recognize the importance of research projects with well-structured study designs, defined data-processing steps, and transparent analysis plans (e.g., statistical analysis plans) [115,116]. For imaging data, such as radiologic images, analysis pipelines have been shown to be suitable to structure the analysis pathway [117]. This is specifically important for big data analysis where interdisciplinarity and collaboration become increasingly important. The development and use of statistical and reporting guidelines support researchers in making their projects more reproducible [118].

Transparency in all the study-design steps (i.e., from hypothesis generation to availability of collected data and code) is specifically relevant for public health and epidemiological research in order to encourage funding agencies, the public, and other researchers and relevant stakeholders to trust research results [119]. Similarly, as globalization and digitalization increase the diffusion of infectious diseases [120] and behavioural risks [121], research practices that foster reproducible results are imperative to implement and diffuse interventions more swiftly.

We believe that these recommendations outlined in this commentary are not unique to big data and that the entire research community could benefit from the use of these approaches [122,123,124,125,126,127]. However, what has been detailed in this commentary is specifically pertinent for big data, as an increase in the volume and complexity of data produced requires more structure and consequent data handling to avoid research waste. With clearly defined and openly shared guidelines, we may strengthen the quality of current research and initiate a shift on multiple levels: at the individual level, the institutional level, and the international level. Some challenges are to be expected, particularly when it comes to finding the right incentives for these changes to stick, but we are confident that with the right effort, we can put scientific integrity back at the forefront of researchers’ minds and, ultimately, strengthen the trust of the population in public-health research and, specifically, public-health research leveraging big data for urban public health and digital epidemiology.

The timely implementation of these solutions is highly relevant, not only to ensure the quality of research and scientific output, but also to potentially allow for the use of data sources that originated without public health in mind, spanning various fields that are relevant to urban public health and digital epidemiology. As outlined in this commentary, such data can originate from multiple sources, such as social media, mobile technologies, urban sensors, and GIS, to mention a few. As such data sources grow and become more readily available, it is important for researchers and the scientific community to be prepared to use these valuable and diverse data sources in innovative ways to advance research and practice. This would allow for the expanded use of big data to inform evidence-based decision making to positively impact public health.

## Figures and Tables

**Figure 1 ijerph-20-01473-f001:**
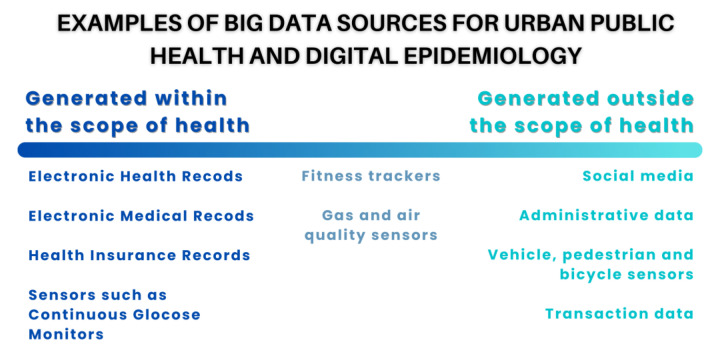
Data sources used in urban public health and digital epidemiology research can broadly be organized along a continuum of health orientation of the process that generated them.

**Figure 2 ijerph-20-01473-f002:**
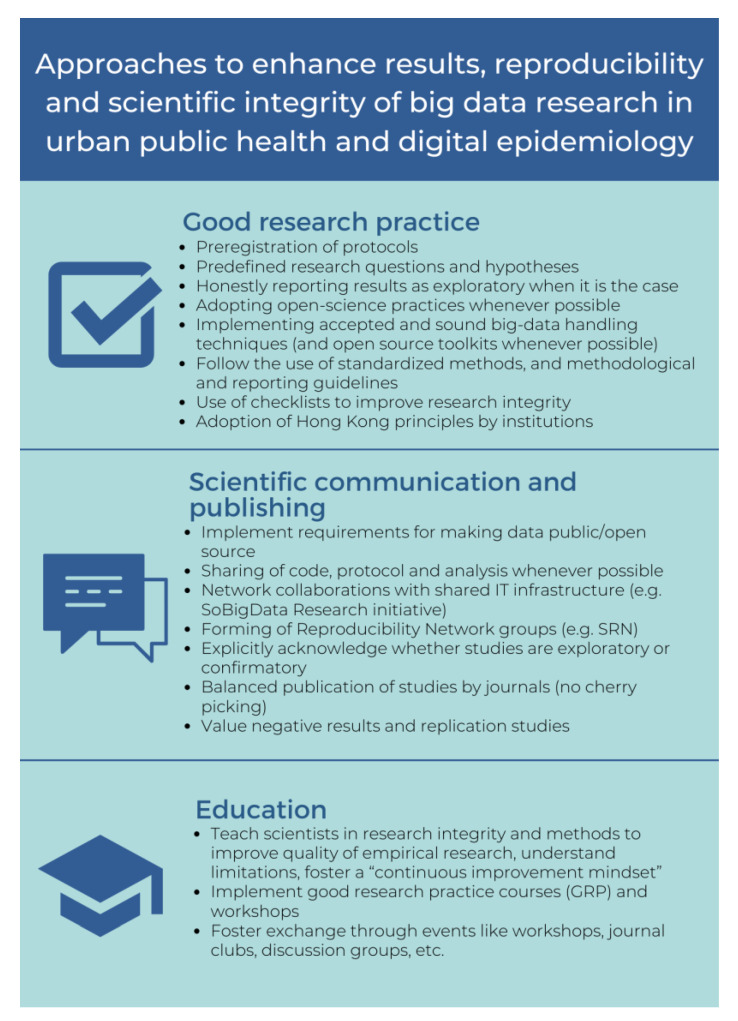
Approaches that address good research practice, scientific communication, and education are important to improve reproducibility and scientific integrity.

**Figure 3 ijerph-20-01473-f003:**
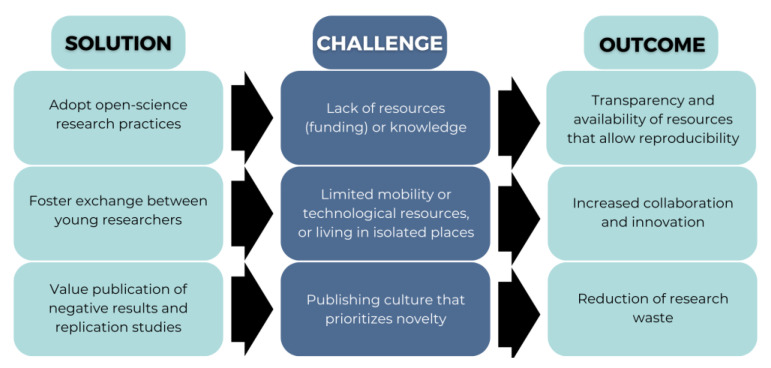
Examples of challenges to expect when implementing approaches aimed at improving reproducibility.

## Data Availability

Not applicable.

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
