# Peer review of "Reproducibility and Scientific Integrity of Big Data Research in Urban Public Health and Digital Epidemiology: A Call to Action"

_ijerph, 2023, doi:10.3390/ijerph20021473_

Round 1
Reviewer 1 Report
I would like to thank you for the opportunity to review the manuscript "Approaches for improving the reproducibility and scientific in-2 tegrity of big data research in urban public health and digital epidemiology". The paper addresses a relevant topic such as the use of big data techniques in the field of public health.
Here are some ideas to help authors improve their work:
1. The references are quite up to date. However, it is not clear to me what type of study they have chosen. If it is a systematic review study the authors would have to refer to the methodology followed and of course apply PRISMA.
2. Authors should clearly state the aim of their research and clearly define the research objectives and hypotheses. Even if it is a systematic review, it should be done in this way.
3. The authors should include some figures and tables to help the reader follow the work and the conclusions of the literature review.
4. Authors have to check that the citations in the reference appendix do not follow the rules of the journal.
5. Authors in the text should review the way in which authors' citations are included. In MPDI, names and years are not given but numbers (e.g. line 4 paragraph 1 Introduction).
Reviewer 2 Report
This paper proposed a approaches to reduce research waste and improve the reproducibility and replicability of big data research in public health, the proposed measure has the room to be improved before the acceptance of the manuscript.
1.The abstract should reflect the contributions of the manuscript. I suggest rewriting it.
2.Keywords must reflect the core of study same as abstract
3. Introduction should be clearly presented to highlight main ideas and motivation behind the
proposed research. Please include and clearly state research question and motivation of proposed
study in Introduction
4. the authors should analyze how to set the parameters of the proposed methods in the framework. Do they have the “optimal” choice?
5.Section experiment, it would be good to have more information about how experiments have been conducted. What tools/software has been used?
6.It will be valuable to provide some analysis or discussion on the computational complexity for the proposed framework.
7.The novelty of this manuscript should be addressed and emphasized in the discussion section.
8.The conclusion section in the present form is relatively weak and should be strengthened with more details and justifications.
9. Figure captions need to be expanded to make them self-explained.
10.The following papers on the same topic should be cited and discussed:
1. Graph Regularized Nonnegative Matrix Factorization for Community Detection in Attributed Networks
2. Diagnosis of Alternaria disease and Leafminer pest on Tomato Leaves using Image Processing Techniques
Reviewer 3 Report
The manuscript entitled: "Approaches for improving the reproducibility and scientific integrity of big data research in urban public health and digital epidemiology" has addressed a very important and influential healthcare problem. Therefore, this paper proposes critical methodological and analytic approaches to reduce research waste and improve the reproducibility and replicability of ample data research in public health. These recommendations are not unique to big data but could benefit the entire research community. With clearly defined and openly shared guidelines, we can improve the quality of current research and initiate change at multiple levels: the individual, institutional, and international levels. Section 2 proposed the solution to the considered problem. Big data research presents a unique opportunity for a cultural shift in how public health research is conducted today. The inherent complexity of working with large data quantities requires a clear and transparent framework. The timely implementation of these proposed solutions is highly relevant as new data sources and methods permeate and become more popular among public health research, including urban public health and digital epidemiology.
Even though the paper provided reasonable solutions, there are many weaknesses in the paper.
1. The paper has technical weaknesses in terms of the complexities of the solutions.
2. There must be case studies because the paper has fewer technical definitions for proposed solutions and problems for healthcare and big data.
3. Anticipated challenges: Table 1. Approaches to enhance results, reproducibility, and scientific integrity of big data research in urban public health. The approaches must be defined very clearly and in detail in the manuscript.
4. Expected outcomes: These should be discussed in more detail. The simulation approaches must be included in the work.
5. The references and existing studies must be included in some papers from the year 2022.
Reviewer 4 Report
1. The title of the paper should be simplified. I suggest a new title: Reproducibility and scientific integrity of big data research in urban public health and digital epidemiology: How to improve it?
2. The goal of the paper should be extended in the introduction. Currently, the paper's goal is written only in three lines: 100-103. It should be extended with more details on what is planned to be done.
3. Instead of jumping to the proposed solution, you should provide an overview chapter of the big data sources in urban public health and digital epidemiology. After that, similar approaches in other fields should be stated. It should be explained in this chapter how your proposed research differs from other proposals for reproducibility and integrity in other fields, especially for big data research.
4. Proposed solution should be in the third chapter. I suggest you put the table at the beginning of the chapter and then elaborate on it. Provide references for Hong Kong standards.
5. I would appreciate it if you create a graphical summary of your proposed approach in which you could include 3 approaches from Table 1, together with the expected outcomes, challenges and implications.
6. At the end of the conclusion, future directions for the research should be provided.
7. I am missing some practical advice on implementing the proposed framework to practice from the paper.
Suggested references are:
Bertoncel, T., Meško, M., & Bach, M. P. (2019, May). Big data for smart factories: A bibliometric analysis. In 2019 42nd International Convention on Information and Communication Technology, Electronics and Microelectronics (MIPRO) (pp. 1261-1265). IEEE.
Pivar, J. (2020). Conceptual Model of Big Data Technologies Adoption in Smart Cities of the European Union. ENTRENOVA-ENTerprise REsearch InNOVAtion, 6(1), 572-585.
Šuštaršič, A., Videmšek, M., Karpljuk, D., Miloloža, I., & Meško, M. (2022). Big Data in Sports: A Bibliometric and Topic Study. Business Systems Research: International journal of the Society for Advancing Innovation and Research in Economy, 13(1), 19-34.
Round 2
Reviewer 1 Report
I congratulate you on the effort you have made to implement the changes. However, if it is a systematic review it should follow the PRISMA model https://www.prisma-statement.org/ . You present a large bibliography (93) but you should focus on citations in the last five years, as this is a highly topical subject in which there are many relevant updated publications. They should also review the way of citing following the journal's rules, and the way of including figures, the titles are too long.Author Response
Please see the attachment.

Reviewer 2 Report
I have gone through the revised paper. All my concerns and requests have been carefully addressed by the authors.
Reviewer 3 Report
The current version of manuscript has meet my all comments. I recommend for acceptance
